# The Effect of Replacing Whole-Plant Corn Silage with Daylily on the Growth Performance, Slaughtering Performance, Muscle Amino Acid Composition, and Blood Composition of Tan Sheep

**DOI:** 10.3390/ani13223493

**Published:** 2023-11-12

**Authors:** Junli Zhang, Fen Li, Rina Na, Xue Bai, Yanfen Ma, Yuwei Yang, Yun Ma, Xiuqin Wang

**Affiliations:** 1Institute of Animal Science, Ningxia Academy of Agriculture and Forestry, Yinchuan 750002, China; zhangjl20000@163.com (J.Z.); yangyuwei0514@163.com (Y.Y.); 2College of Animal Science and Technology, Ningxia University, Yinchuan 750002, China; lifen2843@126.com (F.L.); nrn_99@126.com (R.N.); baixue333work@163.com (X.B.); mayf@nxu.edu.com (Y.M.); mayun@nxu.edu.com (Y.M.)

**Keywords:** daylily silage, tan sheep, growth performance, amino acids, blood biochemical indicators

## Abstract

**Simple Summary:**

To date, few studies in China and abroad have investigated the utilization of daylily. Currently, there is no literature on the evaluation of the feed utilization value of daylily. By using biological fermentation technology, we developed a formula for replacing corn silage with daylily as a feed source for Tan sheep by analyzing coated silage production, diet formulation optimization, growth performance indicators, slaughter performance indicators, carcass quality, and blood biochemical indicators. Based on the feeding test and safety evaluation, the effect of daylily silage on the growth performance and slaughter performance of Tan sheep was evaluated, and the significance of daylily for feed utilization was determined, which provided a scientific basis for further scientific and rational utilization of daylily stems and leaves as feed resources.

**Abstract:**

The shortage of high-quality coarse feed resources is the main factor that restricts the development of animal husbandry in many developing countries. The present study aimed to investigate the effects of replacing corn silage with daylily silage on the growth performance, slaughter performance, blood biochemical indicators, meat quality, and muscle amino acid composition of Tan sheep. A total of 72 healthy Tan sheep were randomly assigned to four groups. In each group, 0%, 20%, 40%, and 60% of corn silage were replaced with daylily silage (denoted as CON, HC20, HC40, and HC60, respectively). Tan sheep fed with daylily silage showed no significant adverse effects on their growth performance, meat quality, and muscle amino acid composition (*p* > 0.05). Some increase was observed in the carcass fat content value (GR-value, *p* < 0.05), thickness of backfat (*p* < 0.05), and the blood urea level (*p* < 0.05). These findings indicate that the utilization of daylily silage instead of whole-plant corn silage has no adverse effects on the growth performance and meat quality of Tan sheep, thus indicating that it can partially replace whole-plant corn feed as a feed resource for Tan sheep.

## 1. Introduction

The shortage of high-quality roughage resources is the main factor that restricts the development of livestock industry in many developing countries [1]. Presently, several studies have reported that locally produced agricultural byproducts or herbaceous plants can replace whole-plant corn feed to some extent and promote the growth and health of livestock and poultry. Liu et al. [2] found that the addition of different proportions of wheat instead of corn in beef cattle diets can improve the apparent digestibility values of dry matter, organic matter, and crude protein and increase serum alanine aminotransferase levels but has no significant effect on rumen pH and fermentation. This finding indicates that an appropriate amount of wheat can be used to replace corn to meet the energy and fiber needs of beef cattle [2]. Silva observed that partial substitution of corn silage (CS) with whole soybean silage (SS) or black oat silage (OS) could reduce the intake and increase the rumination and chewing activities of cows without affecting their growth performance [3]. Zhang et al. reported that sweet straw and wheat straw can be used as cheaper substitutes for whole-plant corn silage (WPCS), without significantly reducing the economic benefits of fattening beef cattle [4]. Natalello et al. found that the addition of whole pomegranate byproducts instead of grain feed to sheep feed improved the antioxidant capacity of sheep meat through the presence of vitamin E in whole pomegranate byproducts [5]. Although many substitutes for WPCS in production have been reported, there is no relevant research on the use of daylily (*Hemerocallis citrina* Baroni) to replace WPCS.

Daylily, a perennial herb from the Liliaceae family, is suitable as both medicine and an edible product. It is named “zheng zhu cai” in China and is usually planted as a forage plant in southwest China. Young daylily stem contains abundant nutrients such as proteins, vitamin C, calcium, fats, carotene, amino acids, and other necessary nutrients beneficial for human health. Daylily is a healthy food with low fat and high protein content, and it is rich in minerals, flavonoids, and phenols, which are considered as active ingredients of daylily [6,7]. Daylily has several important medicinal properties, such as achieving hemostasis and anti-inflammatory effects, clearing heat, removing dampness, helping digestion, improving eyesight, and smoothing the nerves, and it can be used as a postpartum supplement [8]. Daylily can also be used as a high-quality forage food to feed livestock. Therefore, daylily is widely planted in northwest China (Yanchi County and the Hongsipu District of Wuzhong City in Ningxia) and has helped to form a local industry. Currently, there are no investigations on the use of daylily silage to replace WPCS for feeding Tan sheep. Hence, in the present study, we conducted a feeding experiment to determine how the addition of daylily silage to the diet of Tan sheep affects their growth performance, slaughter performance, meat quality, muscle amino acid composition, and blood biochemical indicators.

## 2. Materials and Methods

### 2.1. Crops and Silages

According to the local corn (milk-ripening period) and harvest time of daylily, the silage period is approximately 60 days from 20 September each year. Daylily silage was coated silage self-made by the study team. WPCS was self-made in the experimental field. Daylily and corn were cut into 2–3 cm pieces and placed in silage. A compactor was used to compact the plastic cloth on the back cover. Silage samples were collected from four different points in the silage cellar. Table 1 shows the nutritional components of daylily silage and corn silage feed.

The concentrated feed composition was as follows: soybean meal, rapeseed meal, cottonseed meal, DDGS (distiller’s dried grains with solubles), stone powder, calcium hydrogen phosphate, soybean oil, sodium chloride, various trace elements, vitamins, and amino acids. The confirmed nutrient composition was as follows: crude protein ≥ 33.0%, crude fiber ≤ 18.0%, crude ash ≤ 20.0%, calcium 1.5–5.0%, total phosphorus ≥ 0.5%, sodium chloride 2.5–5.0%, lysine ≥ 1.1%, and moisture ≤ 13.0%.

### 2.2. Animals and Diets

The experiment was conducted at Ningxia Shuomu Yanchi Tan Sheep Breeding Co., Ltd. (Yinchuan, China). Based on random grouping, 72 healthy 3-month-old Tan sheep weighing approximately 24.4 ± 2.4191 kg were randomly assigned to 4 groups. Daylily silage was used to replace 20%, 40%, and 60% corn silage in the diet of Tan sheep as a total mixed diet (TMR). This experiment was conducted in groups, and the animals were fed twice a day (at 8 a.m. and 4 p.m.) after accurately weighing according to Table 2. The nutritional components of the feed are shown in Table 3.

### 2.3. Determination of Sheep Growth Performance and Slaughter Performance

#### 2.3.1. Determination of Sheep Growth Performance

During the experiment, we tracked and tested the growth performance of all 72 Tan sheep (18 sheep in each group). The amount of feed provided and rejected (10% provided) was recorded twice a day, and the animals were weighed to determine the feed intake for each animal. Average daily gain (ADG) and feed conversion rate (FCR) were calculated [9].

#### 2.3.2. Determination of Sheep Slaughter Performance

At the end of the experiment, 3 sheep were randomly selected from each group for slaughter. According to the standard commercial procedure [10], Ministry of Agriculture, People’s Republic of China), sheep were electrocuted and slaughtered humanely in the slaughterhouse. Before slaughtering, the animals were fasted for 24 h and stopped from drinking water for 2 h before weighing; the live weight of the animals was recorded before slaughtering.

Carcass weight: after slaughtering and releasing the blood, the test sheep were skinned, and the whole body (including kidney and surrounding fat) with head, hoof, and visceral tissue removed was weighed after standing for 30 min [11].

Net meat weight and bone weight: the weight of the meat after all bones were removed from the carcass was defined as net meat weight; the weight of the collected bones was defined as bone weight.

GR value: We measured the tissue thickness between the 12th and 13th ribs of a sheep carcass at a distance of 11 cm from the midline of the spine. We used this as an indicator to evaluate the fat content of the Tan sheep carcass. A vernier caliper (Booher-303215, Kunshan Kaipai Hardware, and Electrical Co., Ltd., Jiangsu, China.) was used to measure issue thickness.

Backfat thickness: A vernier caliper was used to measure the fat thickness just above the middle of the eye muscle between the 12th and 13th pairs of ribs.

Eye muscle area: The cross-sectional area of the longissimus dorsi muscle (LDM) at the interface of the 12th and 13th ribs was measured on a sulfur paper by using a planimeter (kp-21c, Koizumi Co., Ltd., Tsuchiura, Japan).
Slaughter rate = (carcass weight/live weight before slaughter) × 100%
Net meat rate = (net meat mass/carcass mass) × 100%
Meat-to-bone ratio = (carcass net meat weight/carcass bone weight) × 100%

#### 2.3.3. Determination of Sheep Meat Quality

In the experiment, we slaughtered 3 Tan sheep in each group. We collected approximately 200 g of LDM samples from the left side of the 12th rib of the slaughtered Tan sheep. Then, we divided them into three sub samples to evaluate meat quality. The quality of meat was tested by measuring muscle water retention capacity (WRC), cooking loss, and shear force. Cooking loss and shear force were determined according to the method of Honikel (1998) [12].

The LDM sample was cooked in a temperature-controlled water bath in a plastic bag to an internal temperature of 75 °C, and the internal temperature of the sample was measured using a digital thermometer. The sample was weighed before and after cooking. Cooking loss was calculated based on the weight difference between raw and cooked samples and expressed as a percentage of the initial weight.

After removing the fat attached to the surface, the LDM was aged in a refrigerator at 4 °C for 24 h. Subsequently, the meat sample was kept at room temperature for 1 h, a thermometer was inserted into the center of the meat sample, and the sample was heated in an 80 °C constant temperature water bath until the temperature reached 70 °C. The meat sample was then removed, cooled in a refrigerator at 0~4 °C, and cut vertically in the direction of muscle fibers to obtain 50 cm-thick meat slices with a diameter of 1. A 27 cm circular sampler (SM-8007 type meat shear force measuring instrument, Dongguan Licheng Electronic Technology Co., Ltd., Dongguan, China) was used to sample and measure the shear force of the meat product along the direction of muscle fibers, and an average value was obtained through multiple measurements. The shear force was expressed in Newtons (N).

WRC was determined using the method described by Franco et al. [13]. Briefly, 10 g of meat samples was wrapped in double-layer gauze, and 18 layers of filter papers were placed on the top and bottom of the meat samples. The samples were then subjected to a pressure of 35 kg for 5 min and weighed immediately. WRC was calculated as follows: WRC = ((total muscle weight − muscle loss)/total muscle moisture) × 100%.

### 2.4. Determination of Amino Acid Composition

In the experiment, we collected approximately 100 g LDM samples from the left side of the 12th rib of the 3 slaughtered Tan sheep in each group. Then, we divided them into three sub samples to measure the amino acid composition of the meat. The content of various amino acids in the carcass was determined using an amino acid auto-analyzer. Before the experiment, excess oil was first removed, and the meat was then cut into small particles. The meat was then ground into a puree by using a food grinder, frozen, and stored. The amino acid composition was determined according to the method described in the national standard [14]. 

The amino acid composition of the LDM samples was determined using a high-performance liquid chromatography system (Waters 2695 HPLC system, Waters Technology, Taunton, MA, USA). Approximately 100 mg of minced muscle samples was hydrolyzed with 20 mL of HCl (6 mol/L) at 110 °C for 22 h in sealed evacuated tubes. The samples were then subjected to automatic precolumn derivatization. After derivatization, 10 µL of each sample was injected into the HPLC system to determine amino acid composition. Mobile phase A was 40 mM NaH_2_PO_4_ adjusted to pH 7.8 with NaOH, while mobile phase B was 45% acetonitrile, 45% methanol, and 10% deionized water. The chromatographic column temperature was set at 37 °C with a flow rate of 1 mL/min. The identity and quantity of the amino acids were assessed by comparison with the retention times and peak areas of standard amino acids.

### 2.5. Determination of Blood Biochemical Indicators

On the morning of the last day of the experiment, 10 sheep were randomly selected from each group, and 5 mL of fasting jugular vein blood was collected. The blood sample was left to stand for 30 min and then centrifuged at 2200× *g* rpm for 10 min to precipitate serum. The contents of total protein (TP), albumin (ALB), aspartate aminotransferase (AST), alanine transaminase (ALT), blood glucose (GLU), urea (UREA), creatinine (CREA), creatine kinase (CK), creatine kinase isoenzymes (CK-MB), lactate dehydrogenase (LDH), and cholinesterase (CHE) in serum were determined using Mindray BS-420 and Hwld DR-200BS automatic biochemical analyzer at Beijing Huaying Biotechnology Research Institute (Beijing, China).

### 2.6. Data Analysis

All data were processed using Microsoft Excel and analyzed using SAS version 9.3 (SAS Institute Inc., Cary, NC, USA). ANOVA and a graph-based test were used to analyze significant differences between the treatments. A *p*-value of <0.05 was considered statistically significant. Data are expressed as mean ± standard error.

## 3. Results

### 3.1. Effect of the Addition of Different Proportions of Daylily Silage on the Growth Performance of Tan Sheep

Table 4 shows the effect of the addition of different proportions of daylily silage instead of corn silage on the growth performance of Tan sheep. The HC20 group showed the highest final weight, and the weight was increased by 1.88%, 1.28%, and 4.10% as compared to those of the CON, HC40, and HC60 groups, respectively. The lowest FCR was noted for the HC20 group, and the FCR value for this group decreased by 11.02%, 12.27%, and 13.15% as compared to those for the CON, HC40, and HC60 groups, respectively. However, the initial weight, final weight, average daily gain, and feed-to-weight ratio were not significantly different between the groups (*p* > 0.05).

### 3.2. Effect of the Addition of Different Proportions of Daylily Silage on the Slaughter Performance of Tan Sheep

Table 5 shows the results of the slaughter performance analysis. No significant differences were noted in live weight, carcass weight, net meat weight, bone weight, fat thickness, and eye muscle area among the experimental groups. However, the GR values of the HC20 and HC40 groups were significantly higher than that of the CON group (*p* < 0.05). The backfat thickness of sheep fed with HC40 and HC60 daylily silage was also significantly lower than those fed with 100% WPCS (*p* < 0.05). Compared to the CON group, the net meat weight of the HC20, HC40, and HC60 groups was 6.57%, 4.85%, and 1.72% higher, respectively, and the net meat percentage of these groups was 8.26%, 6.45%, and 1.69% higher than that of the CON group, respectively.

### 3.3. Effect of the Addition of Different Proportions of Daylily Silage on the Meat Quality of Tan Sheep

Table 6 shows the results for the meat quality analysis. The HC40 group showed the highest water loss rate, which was 3.51%, 3.86%, and 9.07% higher than those of the CON, HC20, and HC60 groups, respectively. The HC40 group exhibited the highest cooked meat rate, which was 0.77%, 1.39%, and 0.92% higher than those of the CON, HC20, and HC60 groups, respectively. No significant difference in shear force values was observed among the groups.

### 3.4. Effect of the Addition of Different Proportions of Daylily Silage on Amino Acid Composition of Tan Sheep Meat

Seventeen types of amino acids were determined in the meat of each group. Table 7 shows the results of amino acid composition in the meat. There were 8 types of essential amino acids, and the total amino acid content reached 44.31%, 44.42%, 44.23%, and 44.11% for the CON, HC20, HC40, and HC60 groups, respectively. Semi-essential amino acids (conditionally essential amino acids) are those amino acids that can be synthesized in the body, but usually cannot meet normal needs and require additional supplements. Three types of semi-essential amino acids were detected in the meat, and their content reached 14.77%, 14.68%, 14.73%, and 14.78% in the CON, HC20, HC40, and HC60 groups, respectively; however, none of them reached a significant level. Six types of non-essential amino acids were detected in the meat, and their content was 40.92%, 40.90%, 41.03%, and 41.11% in the CON, HC20, HC40, and HC60 groups, respectively.

The taste of amino acids is closely associated with the hydrophobicity of their side chain R groups. Amino acids with low hydrophobicity mainly have a sweet taste and are termed sweet amino acids, such as glycine, alanine, serine, lysine, and aspartate. Amino acids with high hydrophobicity mainly have a bitter taste and are termed bitter amino acids, such as leucine, isoleucine, phenylalanine, tyrosine, tryptophan, histidine, lysine, and arginine. For amino acids with an acidic side chain R group (such as COOH and SO_3_H), the main flavor is acidity, such as aspartate and glutamate. These two amino acids are also important prerequisites for the formation of delicious substances and are also known as umami amino acids. The ANOVA analysis showed that their content and proportion were not significantly different among the different groups of umami amino acids, sweet amino acids, bitter amino acids, and total amino acids (Table 8).

### 3.5. Effect of the Addition of Different Proportions of Daylily Silage on Blood Biochemical Indicators of Tan Sheep

Table 9 shows the effect of the addition of different proportions of daylily silage instead of corn silage on the blood biochemical indicators of Tan sheep. The results showed that the content of UREA in the HC60 group was significantly higher than that in the CON group (*p* < 0.05). The groups showed no significant differences in the levels of TP, ALB, AST, ALT, CREA, LDH, CHE, CK, CK-MB, and GLU.

## 4. Discussion

Daylily silage and WPCS showed differences in their nutritional composition. Daylily silage showed higher CP, DM, EE, CF, NDF, and ADF content, but lower Ga and *p* content. Zhang et al. reported a positive correlation between the CP content in feed and body weight, ADG, and dry matter intake in ruminants [15]. Some studies have also shown that the protein content of feed can significantly affect the dietary intake, health, and growth performance of livestock [1,16,17,18]. The growth performance of sheep in the HC40 and HC60 groups was not as high as that in the HC20 group, particularly for the HC60 group. This might be due to the poor quality of silage caused by excessive CP in HC40 and HC60 silage. Previous studies have shown that forage feed with a high protein content is not suitable for producing high-quality silage because of the high buffer capacity, pH value, and extensive proteolysis during ensiling [19,20].

Carcass trait is a key indicator that reflects the ideal livestock production in animal husbandry [3,21], which is closely associated with feed composition [22]. The experimental results showed that the use of daylily silage instead of 40% WPCS to feed sheep had higher GR values and lower backfat thickness (*p* < 0.05). The results showed that the body fat content of Tan sheep fed with Daylily silage increased, while the subcutaneous fat accumulation was less, with more edible parts. The slaughter performance of Tan sheep fed with daylily silage was better than that of Tan sheep fed with corn silage. This might be due to the reasonable balance of CP and ME concentrations in the diet. Wang et al. found that as the content of sweet sorghum silage feed increased, the CP content in the diet increased and the ME level decreased (which is consistent with the trend of dietary proportion in this experiment), resulting in less subcutaneous fat accumulation in Kurar sheep [1]. This finding was consistent with the results of the present study. The shear force of HC20, HC40, and HC60 lambs was smaller (*p* < 0.05), which may be attributed to the higher intramuscular fat deposition in the carcass. The increase in intramuscular fat will reduce the physical strength of muscle fibers; thus, making it easier for muscle bundles to separate. Consequently, muscles become softer and easier for consumers to chew. Intramuscular fat helps improve flavor, thereby increasing consumer acceptance of the product [23]. Previous studies have shown that lower shear forces can also be obtained through higher protein content [24] and appropriate energy levels [25] in the diet.

Amino acids are the most important nutrients in meat, and their content and composition are important indicators to evaluate the physiological value of food proteins [26]. The present study showed the presence of 17 types of total amino acids, and their content was balanced, with no significant difference. The ratio of essential amino acids to total amino acids in each group was above 44%. The contents of umami, sweet, and bitter amino acids in the meat were slightly different in each group; however, the differences among the groups were not significant. Ningxia meat is a delicious and nourishing food, and its glutamic acid and aspartic acid contents account for more than 25% of the total amino acid content [27]; this finding is consistent with the results of the present study. The content of various amino acids in the experimental sheep was generally relatively balanced, thus providing meat with rich and balanced amino acid content in each group.

Daylily contains various bioactive components, among which flavonoids and phenols are considered the active ingredients [6]. Flavonoids have anti-inflammatory, antiviral, and anticancer properties [28]. Phenolic compounds show potential antioxidant activity [29]. Although the present study did not explicitly explore the effects of daylily feed on the antioxidant and inflammatory properties in the meat of Tan sheep, the results confirmed that feeding daylily feed had no harmful effects on the health of Tan sheep. The normal levels of serum total protein, albumin, and globulin in sheep are 60–70 g/L, 24–30 g/L, and 35–57 g/L, respectively [30]. In the present study, all serum indices in the experimental groups were within the normal range. High serum total protein concentration is a manifestation of vigorous protein metabolism, which is conducive to promote animal growth and improve FCR. The serum UREA nitrogen content reflects the protein metabolism of the animal body and more accurately reflect the balance of animal protein metabolism and dietary amino acids [31,32]. The CP level of the experimental diet increased slightly following the addition of daylily silage, and the blood urine nitrogen content increased with the increase in daylily silage content in the diet. The serum UREA nitrogen content in the HC60 group was significantly higher than that in the CON, HC20, and HC40 groups. The substitution of WPCS with HC60 containing daylily silage significantly reduced the protein utilization rate of Tan sheep. These results indicated that the addition of 60% daylily silage to WPCS was not conducive to protein decomposition in Tan sheep. Enzyme activities in serum could be used as a determinant of various pathological diseases. Aminotransferase activities of the AST and ALT enzymes in sera are the key determinants for hepatocellular damage [33]. ALT and AST, the measures of liver function, are elevated in hepatocyte necrosis or alcoholism. In the present study, no significant change was observed in the AST level, and the AST content of the groups fed with daylily silage was reduced after WPCS was replaced. This finding indicated that feeding daylily silage had no adverse effect on the liver of Tan sheep. 

## 5. Conclusions

The results of the present study suggest that replacing corn silage with daylily silage in the diet of Tan sheep has no negative effects on growth performance, slaughter performance, meat quality, amino acid composition, and blood biochemical indicators. Replacing a portion of corn silage with daylily silage can reduce the backfat thickness of Tan sheep, increase the GR value, and reduce shear force. Additionally, the serum urea content significantly increased. Thus, it is feasible to feed Tan sheep with dietary silage; however, our experiment only conducted preliminary exploration on small samples. In the future, it will be necessary to further explore the possibility of using Daylily silage feed to Tan sheep in large-scale experiments. The present study provides a theoretical basis for feeding fattening Tan sheep with daylily silage feed in the future.

## Figures and Tables

**Table 1 animals-13-03493-t001:** Nutrient composition of daylily silage and corn silage.

Items	Daylily Silage	Corn Silage
Crude protein (CP)	7.38%	6.04%,
Dry matter (DM)	31.7%	29.7%
Crude ash	13.1%	8.2%
Crude fat (EE)	2.34%	1.9%
Crude fiber (CF)	40.4%	24.8%
Neutral detergent fiber (NDF)	57.5%	51.0%
Acidity detergent fiber (ADF)	54.2%,	27.4%
Calcium	1.32%	2.80%
Phosphorus	0.12%	1.76%
pH	4.06	2.85
Ammoniacal nitrogen	0.21 g/kg	0.39 g/kg
Lactic acid	1.52 g/100 g	0.704 g/100 g
Acetic acid	2.94 g/kg	11.17 g/kg
Isobutyric acid	0.12 g/kg	0.12 g/kg
Butyrate	0.158 g/kg	0.025 g/kg

**Table 2 animals-13-03493-t002:** Daylily feed ingredient addition for experimental sheep.

	Component (kg)	CON	HC20	HC40	HC60
Concentrate composition	Corn	8.6	8.6	8.6	8.6
	Concentrated material	2.5	2.5	2.5	2.5
Forage composition	Daylily silage	0	3.2	6.4	9.6
	Corn straw	0.8	0.8	0.8	0.8
	Corn silage	16	12.8	9.6	6.4
	Alfalfa straw	0.8	0.8	0.8	0.8
	Total (kg/d)	28.7	28.7	28.7	28.7

Notes: CON, HC20, HC40, and HC60 represent the replacement of 0, 20, 40, and 60% of corn silage with Daylily silage, respectively.

**Table 3 animals-13-03493-t003:** Nutrient composition of the feed given to the four groups.

Raw Material	Abbreviation	CON	HC20	HC40	HC60
Dry matter	DM%	74.87	74.76	74.64	74.53
Crude protein	CP%	13.96	14.51	15.07	15.62
Digestible energy	DE (MJ/kg)	11.48	11.05	10.62	10.19
Metabolizable energy	ME (MJ/kg)	9.41	9.03	8.66	8.28
Neutral detergent fiber	NDF%	22.78	22.96	23.13	23.31
Acid detergent fiber	ADF%	13.03	14.14	15.25	16.36
Calcium	Ga%	0.24	0.32	0.34	0.35
Phosphorus	P%	0.23	0.23	0.24	0.25

Notes: DE = total energy (GE)—fecal energy (FE); ME = DE—urinary energy (UE)—gas energy (Eg). CON, HC20, HC40, and HC60 represent the replacement of 0, 20, 40, and 60% of corn silage with Daylily silage, respectively.

**Table 4 animals-13-03493-t004:** Effect of daylily silage addition on the growth performance of Tan sheep.

Items	Treatments				SEM	*p*
	CON	HC20	HC40	HC60		
Initial weight (kg)	25.41 ± 1.71	25.18 ± 1.98	25.69 ± 1.95	24.87 ± 2.26	0.301	0.684
Final weight (kg)	35.60 ± 2.75	36.27 ± 2.54	35.81 ± 3.31	34.84 ± 3.44	0.516	0.604
Total weight gain (kg)	10.19 ± 1.99	11.09 ± 0.56	10.12 ± 1.36	9.97 ± 1.17	0.004	0.321
Average daily gain (kg)	0.17 ± 0.04	0.17 ± 0.03	0.16 ± 0.04	0.17 ± 0.03	0.265	0.338
Average daily feed intake (kg/day DM)	1.59 ± 0.01	1.60 ± 0.02	1.61 ± 0.02	1.62 ± 0.02	0.018	0.278
FCR (%)	4.90 ± 1.03	4.36 ± 0.40	4.97 ± 1.11	5.02 ± 1.16	0.439	0.211

Notes: CON, HC20, HC40, and HC60 represent the replacement of 0, 20, 40, and 60% of corn silage with Daylily silage, respectively. SEM represents standard error of mean, n = 18.

**Table 5 animals-13-03493-t005:** Effect of daylily silage addition on slaughter performance of Tan sheep.

Items	Treatments				SEM	*p*
	CON	HC20	HC40	HC60		
Pre-slaughter live weight (kg)	37.37 ± 0.15	37.07 ± 9.05	36.60 ± 0.44	37.83 ± 0.21	0.448	0.262
Carcass weight (kg)	21.00 ± 0.78	20.63 ± 7.58	20.87 ± 0.84	20.53 ± 0.75	0.187	0.859
Net meat weight (kg)	12.78 ± 0.43	13.62 ± 2.72	13.40 ± 0.38	13.00 ± 0.20	0.329	0.379
Bone weight (kg)	3.82 ± 0.34	3.72 ± 0.40	3.42 ± 0.21	4.18 ± 0.15	0.271	0.363
GR value (mm)	18.29 ± 2.15 ^b^	22.86 ± 2.40 ^a^	24.66 ± 3.44 ^a^	18.73 ± 2.64 ^ab^	2.706	<0.001
Thickness of backfat (mm)	7.35 ± 1.67 ^a^	6.22 ± 0.88 ^ab^	5.33 ± 1.08 ^b^	4.78 ± 1.24 ^b^	0.972	<0.001
Eye muscle area (cm^2^)	12.32 ± 1.03	14.55 ± 1.42	15.73 ± 2.53	12.85 ± 0.89	1.357	0.088
Slaughter rate (%)	56.20 ± 1.93	55.75 ± 25.85	57.03 ± 2.67	54.27 ± 1.67	1.002	0.624
Net meat rate (%)	61.40 ± 3.39	66.47 ± 29.04	65.63 ± 1.99	63.79 ± 0.72	1.946	0.091
Meat to bone ratio (%)	3.43 ± 0.75	3.73 ± 1.18	4.00 ± 0.25	3.12 ± 0.31	0.329	0.325

Notes: CON, HC20, HC40, and HC60 represent the replacement of 0, 20, 40, and 60% of corn silage with Daylily silage, respectively. In the same row, mean values with different superscripts indicate significant differences (*p* < 0.05). SEM represents standard error of mean, n = 3.

**Table 6 animals-13-03493-t006:** Effect of daylily silage addition on the carcass quality of Tan sheep.

Items	Treatments				SEM	*p*
	CON	HC20	HC40	HC60		
Water loss (%)	31.94 ± 1.94	31.83 ± 2.11	33.06 ± 2.59	30.31 ± 4.57	1.125	0.268
Cooked meat rate (%)	52.78 ± 1.29	52.42 ± 1.84	53.19 ± 0.30	52.70 ± 1.32	0.318	0.085
Shear force (N)	66.71 ± 2.34 ^a^	52.42 ± 1.51 ^b^	61.34 ± 2.18 ^a^	52.69 ± 1.08 ^b^	5.024	<0.001

Notes: CON, HC20, HC40, and HC60 represent the replacement of 0, 20, 40, and 60% of corn silage with Daylily silage, respectively. In the same row, mean values with different superscripts indicate significant differences (*p* < 0.05). SEM represents standard error of mean, n = 3.

**Table 7 animals-13-03493-t007:** Effect of daylily silage addition on the amino acid composition of Tan sheep.

Types of Amino Acids		Treatments				SEM	*p*
		CON	HC20	HC40	HC60		
Essential amino acids	Threonine	0.83 ± 0.02	0.79 ± 0.03	0.77 ± 0.05	0.77 ± 0.01	0.024	0.275
	Valine	0.93 ± 0.03	0.87 ± 0.03	0.86 ± 0.06	0.87 ± 0.00	0.028	0.209
	Isoleucine	0.85 ± 0.02	0.83 ± 0.03	0.78 ± 0.06	0.81 ± 0.00	0.026	0.239
	Leucine	1.46 ± 0.04	1.39 ± 0.04	1.36 ± 0.10	1.38 ± 0.00	0.038	0.309
	Phenylalanine	0.85 ± 0.03	0.81 ± 0.04	0.79 ± 0.06	0.78 ± 0.00	0.027	0.345
	Lysine	1.64 ± 0.05	1.56 ± 0.06	1.52 ± 0.11	1.53 ± 0.01	0.047	0.232
	Histidine	0.62 ± 0.04	0.58 ± 0.05	0.58 ± 0.05	0.55 ± 0.01	0.025	0.332
	Methionine	0.51 ± 0.01	0.49 ± 0.01	0.46 ± 0.03	0.48 ± 0.00	0.018	0.109
	Subtotal	7.72 ± 0.01	7.36 ± 0.02	7.16 ± 0.03	7.22 ± 0.00	0.217	0.271
Semi-essential amino acid	Arginine	1.14 ± 0.02	1.08 ± 0.03	1.06 ± 0.08	1.07 ± 0.01	0.031	0.238
	Glycine	0.77 ± 0.00	0.72 ± 0.02	0.72 ± 0.09	0.73 ± 0.02	0.021	0.186
	Tyrosine	0.64 ± 0.01	0.61 ± 0.02	0.59 ± 0.04	0.61 ± 0.01	0.018	0.289
	Subtotal	2.43 ± 0.05	2.43 ± 0.09	2.39 ± 0.23	2.42 ± 0.06	0.016	0.194
Non-essential amino acids	Aspartate	1.64 ± 0.05	1.56 ± 0.06	1.53 ± 0.11	1.53 ± 0.01	0.045	0.280
	Glutamate	2.92 ± 0.08	2.78 ± 0.10	2.70 ± 0.18	2.76 ± 0.02	0.081	0.552
	Serine	0.71 ± 0.02	0.68 ± 0.02	0.66 ± 0.05	0.66 ± 0.00	0.020	0.262
	Alanine	1.07 ± 0.02	1.01 ± 0.03	1.00 ± 0.07	1.01 ± 0.00	0.028	0.334
	Proline	0.69 ± 0.01	0.66 ± 0.02	0.66 ± 0.06	0.68 ± 0.01	0.013	0.743
	Cystine	0.06 ± 0.00	0.06 ± 0.00	0.06 ± 0.01	0.06 ± 0.01	0.000	0.723
	Subtotal	7.13 ± 0.03	6.77 ± 0.03	6.64 ± 0.60	6.73 ± 0.01	0.186	0.379

Notes: CON, HC20, HC40, and HC60 represent the replacement of 0, 20, 40, and 60% of corn silage with Daylily silage, respectively. SEM represents standard error of mean, n = 3.

**Table 8 animals-13-03493-t008:** Effect of daylily silage addition on the functional amino acid content in Tan sheep muscle.

Groups	Umami Taste Amino Acid		Sweet Amino Acids		Bitter Amino Acid	
	Content (mg/g)	Percentage of Total Amino Acids (%)	Content (mg/g)	Percentage of Total Amino Acids (%)	Content (mg/g)	Percentage of Total Amino Acids (%)
CON	4.58 ± 0.14	26.26 ± 0.06	4.10 ± 0.08	23.52 ± 0.25	8.75 ± 0.30	50.22 ± 0.26
HC20	4.35 ± 0.16	26.25 ± 0.28	3.88 ± 0.14	23.44 ± 0.20	8.33 ± 0.33	50.31 ± 0.48
HC40	4.24 ± 0.30	26.21 ± 0.29	3.84 ± 0.35	23.68 ± 0.30	8.11 ± 0.63	50.11 ± 0.22
HC60	4.30 ± 0.02	26.27 ± 0.17	3.87 ± 0.04	26.68 ± 0.16	8.19 ± 0.02	50.05 ± 0.02
SEM	0.129	0.023	0.104	1.359	0.247	0.100
*p*	0.217		0.296		0.262	

Notes: CON, HC20, HC40, and HC60 represent the replacement of 0, 20, 40, and 60% of corn silage with Daylily silage, respectively. SEM represents standard error of mean, n = 3.

**Table 9 animals-13-03493-t009:** Effect of daylily silage addition on the blood biochemical indicators of Tan sheep.

Items	Treatments				SEM	*p*
	CON	HC20	HC40	HC60		
TP (g/L)	66.06 ± 3.19	65.09 ± 3.87	64.92 ± 2.92	65.14 ± 6.50	0.443	0.173
ALB (g/L)	24.44 ± 2.19	23.12 ± 2.60	22.53 ± 2.40	24.85 ± 2.35	0.945	0.121
AST (µ/L)	94.09 ± 7.79	83.64 ± 12.62	80.94 ± 17.21	93.18 ± 13.95	5.761	0.076
ALT (µ/L)	21.65 ± 3.92	19.50 ± 7.21	18.08 ± 6.03	23.16 ± 6.19	1.950	0.250
GLU (mmol/L)	1.10 ± 0.44	0.95 ± 0.34	1.01 ± 0.42	1.05 ± 0.35	0.055	0.850
UREA (mmol/L)	3.88 ± 1.18 ^b^	4.39 ± 0.96 ^ab^	4.44 ± 0.63 ^ab^	5.77 ± 0.93 ^a^	0.699	<0.001
CREA (µmol/L)	82.68 ± 8.69	80.84 ± 13.47	75.63 ± 8.36	77.01 ± 12.82	2.839	0.462
CK (µ/L)	262.98 ± 70.47	215.28 ± 51.92	262.39 ± 110.31	235.27 ± 53.83	19.997	0.432
CK-MB (µ/L)	119.56 ± 32.49	95.98 ± 12.14	119.77 ± 51.97	104.58 ± 24.03	10.158	0.350
LDH (µ/L)	539.23 ± 58.96	491.69 ± 91.19	495.55 ± 70.89	512.96 ± 42.22	18.756	0.400
CHE (µ/L)	2254.74 ± 262.15	2124.62 ± 448.11	2194.60 ± 309.11	2004.68 ± 348.87	93.016	0.427

Notes: TP, total protein; ALB, albumin; AST, aspartate aminotransferase; ALT, alanine transaminase; GLU, blood glucose; CREA, creatinine; CK, creatine kinase; CK-MB, creatine kinase isoenzymes; LDH, lactate dehydrogenase; CHE, cholinesterase. CON, HC20, HC40, and HC60 represent the replacement of 0, 20, 40, and 60% of corn silage with Daylily silage, respectively. In the same row, mean values with different superscripts indicate significant differences (*p* < 0.05). SEM represents standard error of mean, n = 10.

## Data Availability

All the tables used to support the findings of the current study are included in the article.

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
