# Peer review of "The Effect of Replacing Whole-Plant Corn Silage with Daylily on the Growth Performance, Slaughtering Performance, Muscle Amino Acid Composition, and Blood Composition of Tan Sheep"

_animals, 2023, doi:10.3390/ani13223493_

Round 1
Reviewer 1 Report
Comments and Suggestions for Authors
This study applied Daylily silage to replace the corn silage in the sheep diet. In the present study, the clinic chemistry data, growth performance and meat quality also reported. However, many lost points in this study should addressed and checked seriously. I suggest the researcher with an animal nutrition professional could be involved in the manuscript preparation.
Title:
The “Daily” stem and leaf silage should be corrected to “Daylily”. Furthermore, the “stem and leaf silage” was not described as a special or important point in the manuscript.
Abstract:
All full text of abbreviation items should be shown in the abstract.
L 17: “Daily” should be “Daylily”.
L23-24: The GC was applied to assay the fatty acid, but no data about fatty acid is shown in this manuscript.
L26: The type of analyzer is not required to be shown in the abstract.
L27-28:” ..did not affect the weight gain and feed digestion of Tan sheep…”. However, this study didn’t measure the digestibility of testing diets.
Introduction:
L43-L51: The literature review in this paragraph was about the broiler. However, the major topic of this study related to the forage source replacement in the ruminant diet.
L64-67: Is there any published nutrient composition or chemical assay data about the Daylily? These data could provide the information to select the optimal supplementation level for Daylily.
Materials and Methods
L80-83: The ensilage condition and ensilage period (how many days?) should be indicated. The forage corn harvest time also affects the quality of corn silage.
L84: The pH value, lactate concentrate, and ammonia concentrate of silages should be shown to confirm the silage quality.
L100-103: The testing diet is fed as TMR ?
L99: The “coarse composition” in Table 2 should be replaced by “forage composition”.
L108: The NDF, ADF and NSC (non-structure carbohydrate) data are important assay results for ruminant diet composition. However, no related information was shown in Table 3.
L109: The full text of DE and ME should be shown in the footnote of Table 3.
legends
L112-115: How to collect the intake data in this study? Sheep was fed in the individual pen or group feeding?
L144: How to determine the shear force that is shown in Table 6?
L153: Is there any pretreatment process applied before the amino acid and fatty acid assay? The fatty acid data of mutton was not shown in the result.
L157-162: Creatine, urea, LDH, CHE… data was shown in Table 9, but no assay method or reference was shown.
Result
L180: please show the feed intake data in Table 4.
L196: Table 5. The decimal form should be unified.
L226: Table 7. How to define the semi-essential amino acid? It should be described in the method or result section.
L228: How about the definition of umami, sweet and bitter amino acids?
L235: “There were no significant differences in TP, ALB, AST, ALT, CREA, LDH, CHE, CK, CK-MB and GLU.” The full text of each abbreviation should be shown in the footnote of Table 9. The sample number of each treatment also needs to be shown in the table.
Discussion
L248-249: The author considered that excessive protein leads to lower silage quality. However, this study lacked the ammonia or pH data of silage.
L289-294: According to the sheep clinic chemistry data, the total protein in serum concentrate is normally 60-70 g/L, but the data in Table 9 was much lower than the normal level.
L297-298: “Replacing the whole plant corn silage of HC60 with daylily silage will significantly reduce the protein utilization rate of Tan sheep.” The protein utilization of ruminants should consider the RDP, RUP and MCP (microbial crude protein). The limited protein fraction assay data in this study may not provide enough evidence to infer the conclusion. The protein degradation and synthesis also affect the nitrogen retention of ruminants.
Author Response
RESPONSE TO REVIEWER 1
Dear Editors and Reviewers:
Thank you for your letter and for the reviewers’comments concerning our manuscript entitled “Effect of Replacing Whole-plant Corn Silage with Daylily on the Growth Performance, Slaughtering Performance, Muscle Amino Acid Composition and Blood Composition of Tan Sheep” (ID: animals-2632173).Those comments are all valuable and very helpful for revising and improving our paper, as well as the important guiding significance to our researches. We feel great thanks for your professional review work on our article. Our response is given in normal font and changes/additions to the manuscript are given in the Yellow label.
|
REVIEWER 1 # |
AUTHOR RESPONSE |
PAGE NAUMBER |
|
Title |
||
|
The “Daily” stem and leaf silage should be corrected to “Daylily”. |
We sincerely thank the reviewer for careful reading. As suggested by the reviewer, we have corrected the “Daily” into “Daylily”. |
|
|
Furthermore, the “stem and leaf silage” was not described as a special or important point in the manuscript. |
Thank you for pointing out this point. The manuscript did not include the "stem and leaf silage" as a descriptive focus, so it was deleted in appropriate places. |
|
|
Abstract: |
||
|
All full text of abbreviation items should be shown in the abstract. |
Thank you for pointing this out. We have modified the abstract so that it can be read independently |
PAGE 1 LINE 25-26 |
|
L 17: “Daily” should be “Daylily” |
We sincerely thank the reviewer for careful reading. As suggested by the reviewer, we have corrected the “Daily” into “Daylily”. |
|
|
L23-24: The GC was applied to assay the fatty acid, but no data about fatty acid is shown in this manuscript. |
Fatty acid determination is not included in the manuscript. The content related to fatty acids has been deleted. |
|
|
L26: The type of analyzer is not required to be shown in the abstract. |
The type of analyzer has been deleted. |
|
|
L27-28:” .did not affect the weight gain and feed digestion of Tan sheep…”. However, this study didn’t measure the digestibility of testing diets. |
Thank you for pointing this out. The reviewer is correct. The "feed digestion" has been deleted. |
|
|
Introduction: |
||
|
L43-L51: The literature review in this paragraph was about the broiler. However, the major topic of this study related to the forage source replacement in the ruminant diet. |
Thank you for pointing this out. Unsuitable references have been replaced. |
PAGE 2 LINE 42 |
|
L64-67: Is there any published nutrient composition or chemical assay data about the Daylily? These data could provide the information to select the optimal supplementation level for Daylily.
|
At present, research on the nutritional components of Daylily is very scarce, and it is only known that the active ingredients of Daylily are flavonoids and phenolic compounds. This section has been added to the manuscript. |
PAGE 2 LINE 61-63 |
|
Materials and Methods |
||
|
L80-83: The ensilage condition and ensilage period (how many days?) should be indicated. The forage corn harvest time also affects the quality of corn silage. |
The harvest time for local Daylily and whole plant corn (milk ripening period) is around September 20, 2020, with a silage period of 60 days. This section has been added to the manuscript. |
PAGE 3 Line 76-77 |
|
L84: The pH value, lactate concentrate, and ammonia concentrate of silages should be shown to confirm the silage quality.
|
The pH value, lactate concentrate and ammoniacal nitrogen of silages be shown in Table 1. |
PAGE 3 Table 1 |
|
L100-103: The testing diet is fed as TMR ?
|
The experimental diet belongs to TMR diet. |
PAGE 4 Line 94 |
|
L99: The “coarse composition” in Table 2 should be replaced by “forage composition”. |
The "coarse composition" in Table 2 have been replaced with "forage composition". |
PAGE 4 Table 2 |
|
L108: The NDF, ADF and NSC (non-structure carbohydrate) data are important assay results for ruminant diet composition. However, no related information was shown in Table 3. |
The NDF and ADF data are presented in Table 3. In addition, the dietary formula used in the experimental design was designed according to the meat sheep standard (NYT816-2004), and the NSC indicator was not included in the formula, so it was not included. |
PAGE 5 Table 3 |
|
L109: The full text of DE and ME should be shown in the footnote of Table 3. |
Table 3 has added relevant footnotes. Corresponding footnotes have also been added to all other tables in the manuscript. |
PAGE 5 Table 3 Line 102-103 |
|
legends |
||
|
L112-115: How to collect the intake data in this study? Sheep was fed in the individual pen or group feeding? |
In this experiment, Tan sheep were randomly divided into a control group and an experimental group, and were fed in a full house. Collective feeding is carried out on a group basis, with 18 sheep in each group. They are fed by dedicated personnel and accurately weighed daily before feeding. This section has been added to the manuscript. |
PAGE 5 Line 106-108 |
|
L144: How to determine the shear force that is shown in Table 6? |
The measurement method of shear force is presented in the method of the manuscript. |
PAGE 6 LINE 147-157 |
|
L153: Is there any pretreatment process applied before the amino acid and fatty acid assay? The fatty acid data of mutton was not shown in the result. |
The fatty acid content was not measured in this study, and the relevant content has been deleted from the manuscript. In addition, before using an automatic amino acid analyzer to determine amino acids, we first remove excess oil from the lamb, then cut it into small pieces, use a food grinder to beat it into meat paste, freeze it, and then measure it according to the method in the national standard GB5009.124-2016. |
PAGE 7 Line 165-166 |
|
L157-162: Creatine, urea, LDH, CHE… data was shown in Table 9, but no assay method or reference was shown.
|
The measurement methods for CREA, UREA, LDH and CHE indicators have been added in the Materials and Methods section. |
PAGE 7 Line 186-188 |
|
Result |
||
|
L180: please show the feed intake data in Table 4. |
The average daily feed intake (ADFI) is shown in Table 4 |
PAGE 8 Table 4 |
|
L196: Table 5. The decimal form should be unified. |
The decimal form of Table 5 has been unified. |
PAGE 9 Table 5 |
|
L226: Table 7. How to define the semi-essential amino acid? It should be described in the method or result section. |
The definition of semi essential amino acids has been added to the corresponding section of the results. |
PAGE 10 Line 242-244 |
|
L228: How about the definition of umami, sweet and bitter amino acids? |
The definitions of fresh, sweet, and bitter amino acids have been added to the results section for display. |
PAGE 11-12 Line 255-263 |
|
L235: “There were no significant differences in TP, ALB, AST, ALT, CREA, LDH, CHE, CK, CK-MB and GLU.” The full text of each abbreviation should be shown in the footnote of Table 9. The sample number of each treatment also needs to be shown in the table.
|
Add the full text and sample size for each abbreviation in Table 9. |
PAGE 12 Table 9 |
|
Discussion |
||
|
L248-249: The author considered that excessive protein leads to lower silage quality. However, this study lacked the ammonia or pH data of silage. |
Table 1 has added ammonia nitrogen and pH data for two types of silage feed, making this discussion meaningful. |
PAGE 3-4 Table 1 |
|
L289-294: According to the sheep clinic chemistry data, the total protein in serum concentrate is normally 60-70 g/L, but the data in Table 9 was much lower than the normal level. |
We truly apologize for our careless mistake. We have corrected our experimental data. The corrected serum total protein content is consistent with previous research results (60-70 g/L). Thank you for your reminder. |
PAGE 9 Table 9 |
|
L297-298: “Replacing the whole plant corn silage of HC60 with daylily silage will significantly reduce the protein utilization rate of Tan sheep.” The protein utilization of ruminants should consider the RDP, RUP and MCP (microbial crude protein). The limited protein fraction assay data in this study may not provide enough evidence to infer the conclusion. The protein degradation and synthesis also affect the nitrogen retention of ruminants. |
The inappropriate discussion section has been deleted. |
|
If there are any other modifications we could make, we would like very much to modify them and we really appreciate your help. Thank you very much for your help.
Reviewer 2 Report
Comments and Suggestions for Authors
Effect of Replacing Whole Plant Corn Silage with Daily Stem and Leaf Silage on the Growth Performance, Slaughtering Performance, Muscle Amino Acid Composition, and Blood Composition of Tan Sheep
Review 1
General comments:
The manuscript investigates the effect of partly replacing the corn silage with the daylily silage in the daily meal of Tan lambs on growth performances and meat quality traits. The influence of nutrition and different animals' daily meal composition on growth performance and meat quality has been widely investigated, and today it is considered that animal nutrition, among others, is the most important factor in animal growth performance and meat quality. This research brings the results of the influence of the use of a new nutrient (daylily silage) in the lamb's nutrition and therefore certainly contributes to a better understanding of the mentioned topic.
However, although the manuscript appears to have all the necessary chapters, each of them has significant shortcomings that must be corrected before the work is accepted for publication. The biggest shortcomings were observed in the description of the materials and methods. Also, the results of the research are relatively poorly explained in the discussion, and the comments are often too general and irrelevant to specific results.
Need to check for typographical errors, punctuation, and especially grammar throughout the manuscript.
Specific comments are placed directly in the attached manuscript.

Author Response
RESPONSE TO REVIEWER 2
Dear Editors and Reviewers:
Thank you for your letter and for the reviewers’ comments concerning our manuscript entitled “Effect of Replacing Whole-plant Corn Silage with Daylily on the Growth Performance, Slaughtering Performance, Muscle Amino Acid Composition and Blood Composition of Tan Sheep” (ID: animals-2632173).Those comments are all valuable and very helpful for revising and improving our paper, as well as the important guiding significance to our researches. We feel great thanks for your professional review work on our article. Our response is given in normal font and changes/additions to the manuscript are given in the Blue label.
|
REVIEWER 2 # |
AUTHOR RESPONSE |
PAGE NAUMBER |
|
Title |
||
|
L2 - Are these terms in accordance with the research? |
Thank you for pointing out this point. The manuscript did not include the "stem and leaf silage" as a descriptive focus, so it was deleted in appropriate places. |
PAGE 1
|
|
L5 - The title is too long and awkward. |
Thank you for your suggestion, and we have also taken note of this. The title has been modified. |
PAGE 1 |
|
Simple summary |
||
|
L12 - Simple summary is missing. |
Added Simple summary. |
PAGE 1 Line 10-18 |
|
Abstract |
||
|
L13 - The abstract should be a total of about 200 words maximum. |
The abstract has been reduced to less than 200 words. |
|
|
L14 -16: These sentences are the same as in the Introdution chapter. That should be avoided. Also, it`s not importatnt for the abstract. |
These duplicate sentences have been deleted. |
|
|
L17 - These terms are not in accordance with the terms used in the rest of manuscriplt (especially in the methods). It also applies to the title of the manuscript. |
Thank you for pointing out this point. The manuscript did not include the "stem and leaf silage" as a descriptive focus, so it was deleted in appropriate places. |
|
|
L20- All abbreviations in the abstract should be explained at first mention. This also applies to the rest of abbreviations in the abstract. It should be possible to read and understand the abstract independently, without reading the rest of the manuscript. |
The full text has been added with abbreviations to enable the abstract to be understood and read independently. |
Page 1 Line 25-26 |
|
L24 - Check the English. It also applies to the rest of the manuscript. |
Thank you for your suggestion. The manuscript has been revised by professional English speakers. |
|
|
L27 - Check for typos. In lines 17 and 18 you write these terms with capital letters. Make your writting uniform. |
We were really sorry for our careless mistakes. The manuscript has been revised by professional English speakers. |
|
|
Introduction |
||
|
L41 - References in the text should be placed according the Instruction for Authors. It also aplies to the rest of manuscript. Recommendation: Read the Instruction for Authors. |
The reference format has been modified according to the requirements of the Animals journal. |
|
|
L44 - Reference number is missing. Also, it aplies to the rest references. |
Missing relevant references have been added. |
|
|
L 74 - Check the grammar. |
The manuscript has been revised by professional English speakers. |
|
|
Materials and Methods |
||
|
L99 - Abbreviations should be explained below the table. It also applies to other tables in the text. |
Table 2 has added relevant footnotes. |
PAGE 4 Line 99-100 |
|
L109 - Explain the abbreviations below the table 3. |
Table 3 has added relevant footnotes. Corresponding footnotes have also been added to all other tables in the manuscript. |
PAGE 5 Line 102-103 |
|
L117 - This chapter should be clearly explained. There are a lot of grammatical errors and unclear sentences. |
Thank you for pointing out the point. The method 2.3.2, 2.3.3 and 2.4 has been revised |
PAGE 5 Line 110-135 |
|
Result |
||
|
L171 - These terms should be the same as in the methods (chapters 3.1. and 3.2.). Terminology sholud be uniformed throughout the whole text.
|
Thank you for pointing out this point. The manuscript did not include the "stem and leaf silage" as a descriptive focus, so it was deleted in appropriate places. |
|
|
L176- Were these differences statistically significant? From tha data in the table 4 it seems they were not significant, so the differences found in this case were accidental or it can be said that all group are similar in final weight (so this statement is false). |
These differences are not statistically significant, with p>0.05. The manuscript has also explained them. |
PAGE 8 Line 205-206 |
|
L177- The previous comment also aplies to this claim. These differences can be mentioned (if it is important for the results), but it must be pointed that they were not statisticaly significant. |
These differences are not statistically significant, with p>0.05. The manuscript has also explained them. |
PAGE 8 Line 205-206 |
|
L180- Specify the number of samples per group (n). This should be applied to other tables. |
The corresponding number of samples (n) has been added to all tables in the manuscript. |
PAGE 8 Line 210 |
|
L194- This figure looks nice, but these data are also in the table 5 (not in accordance to instruction for authors). Differences among the 4 groups can be marked by different letters within row. |
Figure 1 has been deleted. Use different letters to display the significance within row in the table 5. |
Page 9 Table 5 |
|
L226- Are there significant differences among groups in the contetnt of essential / semi-esential / non-esential. |
There was no significant difference among groups in the contetnt of essential / semi-esential / non-esential. p>0.05 |
|
|
L238- Same comment as for figure 1. |
Figure 2 has been deleted. Use different letters to display the significance within row in the table 9. |
Page 13 Table 9 |
|
Discussion |
||
|
L255- Thickness of backfat among the groups was also signficantly different (table 5). |
The difference in backfat thickness is also being discussed and the results are increasing. |
PAGE 13 Line 298-300 |
|
L255- According to table 3 the diet of the CON group contained more ME then the other 3 groups. |
Thank you for your opinion. There is indeed a problem with my statement. This section has been revised in the article. |
PAGE 14 Line 303-304 |
|
256-258 It shoud be better explained how research of Wand et al. suports higher GR values in this reserch. |
Thank you for your opinion. Provide a more detailed explanation of Wang et al.'s research in the manuscript. |
PAGE 14 Line 304-308 |
|
L258-259 This sentence is too general for the specific comments made in this paragraph. |
Thank you for pointing out this point. We have had a more specific discussion on the part you questioned. |
PAGE 14 Line 310-315 |
|
L268-272 This information is irrelevant to this research. Maybe it`s for the introduction, not for this chapter. L268-272
|
Thank you for your suggestion. The relevant content has been deleted. |
|
|
L274-277 Too general for the discussion of the results. |
Thank you for pointing out this point. More specific discussions have been added to this section in the manuscript. |
PAGE 14 Line 318-322 |
|
L280- What research? The reference missing. |
Thank you for pointing out this point. Corresponding references have been added. |
PAGE 14 Line 322-324 |
|
L286-288 Too general for the discussion of the concrete results. |
Thank you for your suggestion. More specific discussions have been added to this section in the manuscript. |
PAGE 14-15 Line 332-344 |
|
References |
||
|
L331- Look at the Instruction for the authors |
The reference format has been modified according to the requirements of the Animals journal. |
|
If there are any other modifications we could make, we would like very much to modify them and we really appreciate your help. Thank you very much for your help.
Reviewer 3 Report
Comments and Suggestions for Authors
Overall the manuscript has great merit, since it explores a novel fedstuff, locally sourced. I recommend the manuscript for major revisions only because I believe some important information in missing from materials and methods and discussion.
Abstract: The abstract is well structured but the authors need to add p-values and some of the results.
L79 - Please add the provinience of corn and daylily.
L95 - Please add standard deviation to the initial weight.
L96 - I advise the authors to add the valuesin g/kg of dry matter.
L103 - Please defne "elaborated stuff".
L103 - Specify the conditions.
L120 - Specify the slaughter method.
L145 - Longissimus from which side?
L164 - Were data tested for normality?
L180 - Please present all results with standard deviation.
Discussion: Although the authors cannot compared results with other works using daylily silage since there aren't any, I believe they need to provide some discussion comparing to other silages with similar composition that were used to replace corn silage.
Conclusions: I advise the authors to add either in the conclusions or dicussions, something on anti-nutricional compounds that might yet be found in daylily silages.
Comments on the Quality of English LanguageOnly minor editting of the english language is needed.
Author Response
RESPONSE TO REVIEWER 3
Dear Editors and Reviewers:
Thank you for your letter and for the reviewers’ comments concerning our manuscript entitled “Effect of Replacing Whole-plant Corn Silage with Daylily on the Growth Performance, Slaughtering Performance, Muscle Amino Acid Composition and Blood Composition of Tan Sheep” (ID: animals-2632173).Those comments are all valuable and very helpful for revising and improving our paper, as well as the important guiding significance to our researches. We feel great thanks for your professional review work on our article. Our response is given in normal font and changes/additions to the manuscript are given in the Green label.
|
REVIEWER 3 # |
AUTHOR RESPONSE |
PAGE NAUMBER |
|
Abstract: |
||
|
The abstract is well structured but the authors need to add p-values and some of the results. |
Thank you for pointing out this point. Added p-value and some results in the abstract. |
PAGE 1 LINE 26-29 |
|
Materials and Methods |
||
|
L79 - Please add the provinience of corn and daylily. |
The provinience of two types of silage feed are shown in Table 2. |
PAGE: 4 Table 2 |
|
L95 - Please add standard deviation to the initial weight. |
Thank you for your suggestion. The standard deviation has been added to the initial weight. |
PAGE:4 Line 93 |
|
L96 - I advise the authors to add the valuesin g/kg of dry matter. |
Thank you for your suggestion. In Table 2, we have shown the added amounts of each component of refined feed and coarse feed, but did not calculate the dry matter content of the corresponding components. But we believe that Table 2 can already show our grouping situation. |
PAGE: 4 Table 2 |
|
L103 - Please defne "elaborated stuff". |
It has no practical significance. Deleted. |
|
|
L120 - Specify the slaughter method. |
The slaughter method has been redefined. You can view it in the manuscript. |
PAGE 5 Line 112-116 |
|
L145 - Longissimus from which side? |
In the slaughter experiment, approximately 200g of the longest dorsal muscle was taken from the 12th rib of each experimental sheep's left half carcass, sealed and packaged in a self-sealing bag, labeled, and stored at 0-4℃ for subsequent measurement. This section has been added to the manuscript. |
PAGE 6 Line 137-139 |
|
L164 - Were data tested for normality? |
We conducted a normality test on the data. |
|
|
Result |
||
|
L180 - Please present all results with standard deviation. |
All tables in the manuscript have been modified to include the result of adding standard deviation. |
|
|
Discussion |
||
|
Although the authors cannot compared results with other works using daylily silage since there aren't any, I believe they need to provide some discussion comparing to other silages with similar composition that were used to replace corn silage. |
Thank you for pointing out this point. The corresponding part of the discussion has been added to the manuscript. |
PAGE 14 Line 304-308 |
|
I advise the authors to add either in the conclusions or dicussions, something on anti-nutricional compounds that might yet be found in daylily silages.
|
Thank you for pointing out this point. Our experiment tested the anti- nutritional compounds of Daylily and found that it contains 31.70g/100g of colchicine. However, no colchicine was detected after being made into Daylily silage feed. So this study did not separately discuss Daylily's anti nutritional factors. |
|
If there are any other modifications we could make, we would like very much to modify them and we really appreciate your help. Thank you very much for your help.
Round 2
Reviewer 1 Report
Comments and Suggestions for Authors
The manuscript had improved and suitable to be accepted. However, the number of decimal place about p-value should be checked again in all Tables. ie: p = 0.025 is recommended in AMA guide.
http://amastyleinsider.com/category/statistics/
Author Response
RESPONSE TO REVIEWER 1
Dear Editors and Reviewers:
Thank you for your letter and for the reviewers’ comments concerning our manuscript entitled “Effect of Replacing Whole-plant Corn Silage with Daylily on the Growth Performance, Slaughtering Performance, Muscle Amino Acid Composition and Blood Composition of Tan Sheep” (ID: animals-2632173). Those comments are all valuable and very helpful for revising and improving our paper, as well as the important guiding significance to our researches. We feel great thanks for your professional review work on our article. Our response is given in normal font and changes/additions to the manuscript are given in the Yellow label.
Comments and Suggestions for Authors (REVIEWER 1):
The manuscript had improved and suitable to be accepted. However, the number of decimal place about p-value should be checked again in all Tables. ie: p = 0.025 is recommended in AMA guide.
Reply:
Thank you for your suggestion. We carefully read the AMA Manual of style and modified the decimal places of P-value in the manuscript as required.
If there are any other modifications we could make, we would like very much to modify them and we really appreciate your help. Thank you very much for your help.
Reviewer 2 Report
Comments and Suggestions for Authors
Dear authors,
Studying your article leads to the conclusion that the methods of research are very confusingly explained probably because the study was conducted on a very small sample of animals. First, you state that the study was conducted on 72 lambs, but the growth performance was followed on only 18 lambs (it is not clear why; why not on all 72 lambs?), which is too small a sample to draw relevant conclusions about growth performances. An even smaller sample (n=3; which you do not list in the "Methods", but only below the tables in the "Results") was used to monitor indicators of slaughter value, meat quality, and amino-acid content... Only for blood test indicators the relevant number of blood samples (n=10) were used.
So if the research methods are inadequate, the results are not relevant.
Unfortunately, I have to conclude that the quality of this paper is not sufficient for publication in a journal ranked as Animals.
Comments on the Quality of English Language
The comment on the quality of the English Language is not relevant, since the research methods are not appropriate.
Author Response
RESPONSE TO REVIEWER 2
Dear Editors and Reviewers:
Thank you for your letter and for the reviewers’ comments concerning our manuscript entitled “Effect of Replacing Whole-plant Corn Silage with Daylily on the Growth Performance, Slaughtering Performance, Muscle Amino Acid Composition and Blood Composition of Tan Sheep” (ID: animals-2632173). Those comments are all valuable and very helpful for revising and improving our paper, as well as the important guiding significance to our researches. We feel great thanks for your professional review work on our article. Our response is given in normal font and changes/additions to the manuscript are given in the Blue label.
Comments and Suggestions for Authors (REVIEWER 2):
Studying your article leads to the conclusion that the methods of research are very confusingly explained probably because the study was conducted on a very small sample of animals. First, you state that the study was conducted on 72 lambs, but the growth performance was followed on only 18 lambs (it is not clear why; why not on all 72 lambs?), which is too small a sample to draw relevant conclusions about growth performances. An even smaller sample (n=3; which you do not list in the "Methods", but only below the tables in the "Results") was used to monitor indicators of slaughter value, meat quality, and amino-acid content... Only for blood test indicators the relevant number of blood samples (n=10) were used.
Reply:
Thank you for your suggestion. A total of 72 Tan sheep were divided into four groups with 18 sheep in each group. The growth performance of all Tan sheep was measured in our experiment. 3 sheep in each group were randomly selected for slaughter for slaughter performance, meat quality and muscle amino acid composition determination. In addition, 10 sheep in each group were randomly selected for blood collection for blood biochemical index analysis. The method in the manuscript states the sample size for each group in the experiment. Mark with blue labels in the manuscript. (PAGE:5 Line:111-112; PAGE:7 Line:182-183)
We slaughtered 3 Tan sheep per group due to cost considerations. However, we referred to the literature of Qing et al.1 and found that their slaughter performance measurements only used 3 animals in each group. So I believe that our manuscript experimental data can preliminarily verify our results. In addition, we added the following content to the conclusion of the manuscript: However, our experiment only conducted preliminary exploration on small samples. In the future, it is necessary to further explore the possibility of using Daylily silage feed to Tan sheep in large-scale experiments. (PAGE:15 Line:360-362)
If there are any other modifications we could make, we would like very much to modify them and we really appreciate your help. Thank you very much for your help.
- Li, Q.; Wang, Y.; Tan, L.; Leng, J.; Lu, Q.; Tian, S.; Shao, S.; Duan, C.; Li, W.; Mao, H., Effects of age on slaughter performance and meat quality of Binlangjang male buffalo. Saudi J Biol Sci 2018, 25 (2), 248-252.
Reviewer 3 Report
Comments and Suggestions for Authors
L118: The authors should explicit how many animals were in each group, since it is my understanding that different n were used in different parts of this experiment.
I would advise the authors to present a sentence at least in the conclusions, focusing on the n being small and how these results are preliminary and should be taken as so. Apart from these remarks, I feel my worries have been addressed.
Author Response
RESPONSE TO REVIEWER 3
Dear Editors and Reviewers:
Thank you for your letter and for the reviewers’ comments concerning our manuscript entitled “Effect of Replacing Whole-plant Corn Silage with Daylily on the Growth Performance, Slaughtering Performance, Muscle Amino Acid Composition and Blood Composition of Tan Sheep” (ID: animals-2632173). Those comments are all valuable and very helpful for revising and improving our paper, as well as the important guiding significance to our researches. We feel great thanks for your professional review work on our article. Our response is given in normal font and changes/additions to the manuscript are given in the Green label.
Comments and Suggestions for Authors (REVIEWER 3):
L118: The authors should explicit how many animals were in each group, since it is my understanding that different n were used in different parts of this experiment.
I would advise the authors to present a sentence at least in the conclusions, focusing on the n being small and how these results are preliminary and should be taken as so. Apart from these remarks, I feel my worries have been addressed.
Reply:
Thank you for your suggestion. A total of 72 Tan sheep were divided into four groups with 18 sheep in each group. The growth performance of all Tan sheep was measured in our experiment. 3 sheep in each group were randomly selected for slaughter for slaughter performance, meat quality and muscle amino acid composition determination. In addition, 10 sheep in each group were randomly selected for blood collection for blood biochemical index analysis.
In addition, attention was added to n in the conclusion: However, our experiment only conducted preliminary exploration on small samples. In the future, it is necessary to further explore the possibility of using Daylily silage feed to Tan sheep in large-scale experiments. (PAGE: 15 Line: 360-362)
If there are any other modifications we could make, we would like very much to modify them and we really appreciate your help. Thank you very much for your help.
Round 3
Reviewer 2 Report
Comments and Suggestions for Authors
Dear authors,
I appreciate your efforts to improve the quality of your manuscript, which is now much better. However, there are still minor shortcomings that need to be corrected before publication.
The size of the sample on which the research was carried out, that is, the number of animals that were used to test certain parameters in the manuscript (Chapter 2. Material and Methods) was not transparently explained. Considering that in chapter 2.2. you state that the research was conducted on 72 lambs, and the individual parameters were monitored on a different number of animals, the exact number of animals used for testing the individual parameters should be stated. This refers to chapters 2.3.1, 2.3.2, 2.3.3, 2.4, and 2.5. Otherwise, if it remains as it is, the reader may be misled. Therefore, it is recommended to add a sentence in each mentioned chapter that clearly shows the number of animals per each tested parameter.
Other specific comments:
L 28. All abbreviations in the abstract at first mentioned should be explained.
Best regards,
Reviewer
Author Response
animals-2632173 - 3rd review
RESPONSE TO REVIEWER 2
Dear Editors and Reviewers:
Thank you for your letter and for the reviewers’ comments concerning our manuscript entitled “Effect of Replacing Whole-plant Corn Silage with Daylily on the Growth Performance, Slaughtering Performance, Muscle Amino Acid Composition and Blood Composition of Tan Sheep” (ID: animals-2632173). Those comments are all valuable and very helpful for revising and improving our paper, as well as the important guiding significance to our researches. We feel great thanks for your professional review work on our article. Our response is given in normal font and changes/additions to the manuscript are given in the Blue label.
Comments and Suggestions for Authors (REVIEWER 2)
No. 1: The size of the sample on which the research was carried out, that is, the number of animals that were used to test certain parameters in the manuscript (Chapter 2. Material and Methods) was not transparently explained. Considering that in chapter 2.2. you state that the research was conducted on 72 lambs, and the individual parameters were monitored on a different number of animals, the exact number of animals used for testing the individual parameters should be stated. This refers to chapters 2.3.1, 2.3.2, 2.3.3, 2.4, and 2.5. Otherwise, if it remains as it is, the reader may be misled. Therefore, it is recommended to add a sentence in each mentioned chapter that clearly shows the number of animals per each tested parameter.
Reply 1: Thank you for your advice. We have added the exact number of sheep for testing in chapters 2.3 (2.3.1, 2.3.2, 2.3.3), 2.4 and 2.5 of the manuscript respectively.
2.3.1 Page 5, lines 106-107
2.3.2 Page 5, lines 112-113
2.3.3 Page 6, lines 138-140
2.4 Page 7, lines 166-167
2.5 Page 7, lines 186-187
No. 2: L 28. All abbreviations originally mentioned in the summary should be explained.
Reply 2: Thank you for your reminder. The GR value abbreviations in the summary have been explained so that they can be read and understood independently. (Page 1, line 28)
If there are any other modifications we can make, we would very much like to make them, and we would appreciate your help. Thank you very much for your help.